# Phytochemical Analysis and Molecular Identification of Green Macroalgae *Caulerpa* spp. from Bali, Indonesia

**DOI:** 10.3390/molecules27154879

**Published:** 2022-07-30

**Authors:** I Gede Putu Wirawan, Ni Kadek Emi Sintha Dewi, Maria Malida Vernandes Sasadara, I Gde Nengah Adhilaksman Sunyamurthi, I Made Jawi, I Nyoman Wijaya, Ida Ayu Putri Darmawati, I Ketut Suada, Anak Agung Keswari Krisnandika

**Affiliations:** 1Department of Biotechnology, Faculty of Agriculture, Udayana University, Bali 80361, Indonesia; igpwirawan@unud.ac.id (I.G.P.W.); tezukaemi@gmail.com (N.K.E.S.D.); nyomanwijaya@unud.ac.id (I.N.W.); darmawati@unud.ac.id (I.A.P.D.); ketutsuada@unud.ac.id (I.K.S.); agung_keswari@unud.ac.id (A.A.K.K.); 2Department of Natural Medicine, Universitas Mahasaraswati Denpasar, Denpasar 80233, Indonesia; 3Department of Dermatology and Venereology, Faculty of Medicine, Warmadewa University, Bali 80239, Indonesia; moaixmoai@gmail.com; 4Department of Pharmacology, Faculty of Medicine, Udayana University, Bali 80232, Indonesia; madejawi@unud.ac.id

**Keywords:** DNA barcoding, GC-MS, molecular identification, molecular marker, phylogeny, phytochemical, *tuf*A

## Abstract

The studies of the Bulung Boni and Bulung Anggur (*Caulerpa* spp.) species and secondary metabolites are still very limited. Proper identification will support various aspects, such as cultivation, utilization, and economic interests. Moreover, understanding the secondary metabolites will assist in developing algae-based products. This study aimed to identify these indigenous *Caulerpa* algae and analyze their bioactive components. The *tuf*A sequence was employed as a molecular marker in DNA barcoding, and its bioactive components were identified using the GC-MS method. The phylogenetic tree was generated in MEGA 11 using the maximum likelihood method, and the robustness of the tree was evaluated using bootstrapping with 1000 replicates. This study revealed that Bulung Boni is strongly connected to *Caulerpa cylindracea*. However, Bulung Anggur shows no close relationship to other *Caulerpa* species. GC-MS analysis of ethanolic extracts of Bulung Boni and Bulung Anggur showed the presence of 11 and 13 compounds, respectively. The majority of the compounds found in these algae have been shown to possess biological properties, such as antioxidant, antibacterial, anticancer, anti-inflammation, and antidiabetic. Further study is necessary to compare the data obtained using different molecular markers in DNA barcoding, and to elucidate other undisclosed compounds in these *Caulerpa* algae.

## 1. Introduction

Seaweeds, also known as macroalgae, are eukaryotic and non-flowering plants with no true stem, leaves, or root surrounding their reproductive systems [1,2]. As with terrestrial plants, seaweeds also subsist by photosynthesis. Seaweeds include a variety of pigments and are taxonomically classified into Chlorophyta (green algae that have chlorophyll pigment), Rhodophyta (red algae that have phycocyanin, and phycoerythrin pigments), and Phaeophyta (brown algae that have fucoxanthin pigment) [3].

Seaweed has been consumed for centuries throughout the world, most notably in Asian countries such as Japan, Korea, China, and Indonesia, but on a smaller scale. Seaweeds have also been used for numerous food, fertilizer, and pharmaceutical products [2]. Seaweed species, such as *Caulerpa* spp., *Gracilaria* spp., and *Euchema spinosum,* are utilized and widely consumed in Bali [4,5,6]. *Caulerpa* seaweeds, locally known as Bulung Boni and Bulung Anggur, have a wide range of biological properties. Julyasih et al. [4] discovered that *Caulerpa* sp. possesses the highest antioxidant activity compared to *Gracilaria* spp. and *Euchema spinosum*. Moreover, *Caulerpa* extract could help prevent photoaging through its inhibitory effect on MMP-1 and its protective mechanism against oxidative DNA damage, according to Wiraguna et al. [7].

*Caulerpa* (Bryopsidophyceae) is a multinucleate siphonous coenocyte and its thallus is differentiated into rhizoid, stolon, assimilator, and branchlets (ramuli) [8,9]. The complete tallus of *Caulerpa* seaweed can reach a length of nearly 1 m (e.g., *Caulerpa taxifolia*). Stolons, or green prostrate axis, give rise to photosynthetic fronds that are erect (assimilators). These assimilators can be leaf-like or have a central axis (rachis) with lateral branchlets (ramuli) arranged in a variety of patterns and shapes [10].

Although it has been widely taken and used, to date, no research has identified the taxonomic profile of Bulung Boni and Bulung Anggur. According to Farnsworth et al., [11] species identification is essential to documenting, managing, and sustaining organism diversity. The *Caulerpa* algae are well-known for their great phenotypic plasticity, which refers to the ability of the same species to exhibit a variety of morphological structures in response to changing environmental conditions [12,13]. Accurate identification and characterization of seaweed species are critical for resolving their taxonomic uncertainty, especially for those with commercial interests. Thus, molecular identification is essential to correctly identifying the *Caulerpa* species. DNA barcoding is a precise and rapid molecular technique for identifying species through the use of short genetic markers [14]. The barcode regions *rbc*L, *mat*K, and *tuf*A are frequently employed in green macroalgae. *tuf*A is a chloroplast gene that encodes the elongation factor TU. Numerous studies have utilized *tuf*A as a genetic marker for identifying green algae due to its high rate of amplification and sequencing success [9,15,16,17,18,19,20,21,22,23].

Additionally, the chemical compositions of these *Caulerpa* seaweeds have not been investigated. Identifying the secondary metabolites and their properties in *Caulerpa* algae will assist in their utilization and the production of algae-based cosmetics and pharmaceutical medications. Therefore, it is necessary to study the bioactive components of these two *Caulerpa* algae. The objective of this study was to identify two indigenous *Caulerpa* seaweed species in Bali, Bulung Boni and Bulung Anggur, using *tuf*A sequence as a molecular marker in DNA barcoding, and to analyze their bioactive components using GC-MS analysis.

## 2. Results and Discussion 

### 2.1. Morphological Characterization

Bulung Boni and Bulung Anggur, like other *Caulerpa* algae, have green-colored thalli composed of stolons, rhizoids, and erect fronds or assimilators (Figure 1). Bulung Boni has a distichous assimilator with a maximum height of 10 cm. It also features uncrowded cylindrical or clavate ramuli radially and distichously arranged. On the other hand, Bulung Anggur has an 8 cm irregular assimilator with vesiculated ramuli that have no distinct arrangement. 

*Caulerpa* spp. are known as macroalgae with a high degree of phenotypic plasticity, making precise identification difficult based on morphological observations alone. Morphologically, Bulung Boni resembles *Caulerpa cylindracea*. *C. cylindracea* showed a straightforward morphology, with creeping stolons and spherical branchlets with upright shoots and grape-like ramuli, distributed radially or distichously [10,24,25,26]. In contrast, the morphological characteristics of Bulung Anggur are similar to those of *Caulerpa macrophysa.* Both Bulung Anggur and *Caulerpa macrophysa* have thallus that can grow up to 3–5 m in length and are found in a lower intertidal and upper subtidal area with strong water movement. Colorless rhizoidal holdfasts hold the prostrate terete, bare branching stolon and erect, terete branches of this seaweed in thick clusters on the sandy–muddy substrate [24].

### 2.2. Phytochemical Analysis

Phytochemical analysis using GC-MS resulted in a chromatogram with several peaks. There are 11 compounds in the ethanol extract of Bulung Boni (Figure 2), and 13 compounds in the ethanol extract of Bulung Anggur (Figure 3). Information on chemical names, retention time (Rt), area under curve (AUC), molecular weight, and formula of Bulung Boni and Bulung Anggur are shown in Table 1 and Table 2, respectively. Cyclohexanamine was the dominant compound in Bulung Boni, as shown by the large area under the curve. Meanwhile, Terephthalic acid is the dominant compound in Bulung Anggur. Based on the phytochemical analysis, it can be observed that the ethanolic extracts of Bulung Boni and Bulung Anggur consist mainly of fatty acid compounds.

The pH (7.74–7.92), BOD5/Five-Day Biological Oxygen Demand (2.8–5.4 mg/L calculated using the Delzer and McKenzie standard method [27]), and temperature (28.9–30.5 °C) of the saltwater in the Serangan island region are within the acceptable range for marine biota; however, salinity (29.9–32.7 ppt) is inadequate [28]. Stress generated by salinity can trigger plants and algae to produce higher secondary metabolites [29]. Hence, this condition might affect the growth of Bulung Boni and Bulung Anggur, resulting in the production of several secondary metabolites.

Several compounds with the highest peak in the chromatogram could not be matched to the database library (they have low quality or probability percentage). These compounds could be quite novel and require additional analysis to clarify their nature. However, various compounds in these algae have been proven to have biological activities. Cyclohexanamine is considered the dominant chemical of Bulung Boni. This chemical can be found in several plants, such as *Duddingtonia flagrans* and *Rhazya stricta* [30,31]. *D. flagrans* showed the dominant presence of cyclohexanamine. This plant possesses a potent nematicidal activity [30]. Cyclohexanamine is also present in *Rhazya stricta*, which showed antidiabetic activity by inhibiting various hyperglycemic key enzymes [32]. Eicosane, in Bulung Boni, has been identified in several studies. Eicosane, an alkane compound ordinarily present in wax, acts as the plant protector against physical damage and prevents the plant from dehydration [33,34]. Eicosane possesses anti-inflammatory, antibacterial, and antitumor properties [35,36,37]. The compound 1-Dodecanol is also found in the ethanolic extract of Bulung Boni, which has been proven to have insecticide and antibacterial properties [38,39]. Neophytadiene, present in both Bulung Boni and Bulung Anggur, is a terpene compound with potent antibacterial, antifungal, anti-inflammatory, antioxidant, antipyretic, and analgesic properties [40].

Several researchers have discovered that the majority of the compounds in Bulung Anggur have biological properties. Heptadecane and nonadecane are compounds that have been reported for their antimicrobial activity [36]. Hexadecanoic acid, commonly known as palmitic acid, has antioxidant, pesticide, nematicide, hypo-cholesterolemic [41], and antibacterial properties [42]. Phytol is a chlorophyll constituent that possesses antimicrobial, anticancer, anti-inflammatory, antidiuretic, antidiabetic, and immunostimulatory properties [40]. Hexadecanoic acid and phytol are the dominant chemicals of other Balinese seaweed, Bulung Sangu (*Gracilaria* sp.), which showed an antioxidant and anti-inflammatory activity [5]. Loliolide is a β-carotene derivative that has been extracted from marine algae, such as *Undaria pinnatifida*, *Sargassum crassifolium*, *Corallina pilulifera*, and green alga *Enteromorpha compressa* [43,44,45,46]. Loliolide has anti-cancer, antimicrobial, antioxidant, anti-inflammatory, and antiaging properties, and is also used in treating diabetes and depression [47,48].

TLC (thin layer chromatography) examination revealed the presence of carotenoids, such as β-carotene, Canthaxanthin, Chlorophyll b, Fucoxanthin, Antheraxanthin, Astaxanthin diester, Astaxanthin monoester, and Neoxanthin in Bulung Boni [49]. However, no carotenoid component was detected in Bulung Boni in this investigation. The absence of carotenoid in Bulung Boni might be due to its infinitesimal peak and consequent under-detection in GC-MS analysis. Furthermore, since carotenoid molecules are heat-labile, the GC (gas chromatography) or GC-MS (gas chromatography-mass spectrometry) methods are ineffective for detecting carotenoids [50].

This research establishes that Bulung Boni and Bulung Anggur are chemically distinct. The difference in the number of compounds could be attributable to the environment wherein they grow (temperature, light intensity, salinity, nutrition, and pH), the biotic factors (pathogen or predator), or their genetic structure. Bulung Boni and Bulung Anggur are presumably different variations or species of *Caulerpa* that are genetically distinct and produce different secondary compounds. In GC-MS analysis, ethanolic extracts of *Caulerpa mexicana* and *Caulerpa racemosa* exhibit different compounds; however, these two *Caulerpa* algae have simulant biological activities, such as potent anti-inflammatory, antidiabetic, and antioxidant properties [51,52].

Seaweed, or macroalgae, is an excellent source of protein, polysaccharides, lipids, and fiber. Multiple studies have demonstrated the high lipid content of various green macroalgae, especially *Caulerpa*. In addition, seaweed is also a source of minerals [2]. Further research to investigate protein, amino acids, total sugar, lipid content, and mineral profiles will support the utilization of this *Caulerpa* seaweed.

### 2.3. Molecular Identification

The correct and successful identification of seaweed strains is aided by identification based on comprehensive taxonomy. DNA barcoding is a quick and accurate method. This method is considered the best approach for species identification. DNA barcoding does not require the whole genome sequence but only a short strand of DNA sequences created from the standard marker region of the entire genome [14]. The Bulung Boni and Bulung Anggur were successfully amplified using the *tuf*A primer (Figure 4). The products of PCR amplification of Bulung Boni and Bulung Anggur with *tuf*A primers were 903 and 930 bp, respectively. There were 18 species with an identity percentage of more than 90% (against Bulung Boni and Bulung Anggur) selected from the NCBI-BLAST results and used in the phylogenetic analysis (Table 3). 

The amplification of the *tuf*A sequence performed well in our investigation. The *tuf*A gene is an excellent candidate for a molecular marker due to its conserved nature across various taxa [9]. *tuf*A encodes protein synthesis elongation factor Tu. Tu was moved from the chloroplast to the nucleus within the green algal lineage that gave rise to terrestrial plants [53]. *tuf*A, ITS (internal transcribed spacer), and *rbc*L (rubisco large subunit) molecular markers are frequently used in phylogeny identification. Nevertheless, *tuf*A produced the best results in identifying Chlorophyceae compared to ITS and *rbcL* [54]. Numerous other studies have also demonstrated that *tuf*A has excellent amplification and sequencing results for identifying green algae compared to other markers [16,17,18].

According to the phylogenetic tree reconstruction results (Figure 5), Bulung Boni is closely related to *Caulerpa cylindracea,* with a 100 percent bootstrap value. Thus, morphological and molecular identification of Bulung Boni produced similar findings. Additionally, Bulung Boni and *Caulerpa cylindracea* have a genetic distance of 0.000 (Table 3). *Caulerpa cylindracea* has been raised from *Caulerpa racemosa* var. *cylindracea* because of its genetic independence [12]. *C. cylindracea* is one of the most invasive algae species that has been reported by several researchers [55,56,57,58]. As a result of its invasive character and high productivity, Bulung Boni has the potential to be utilized and commercialized. 

Bulung Anggur, on the other hand, seems to be an outgroup on the phylogenetic tree, indicating that it is genetically distinct from other *Caulerpa* homologs determined by NCBI-BLAST. Bulung Anggur shares morphological traits with *Caulerpa racemosa*, particularly with *Caulerpa racemosa* var. *macrophysa* (Sonder ex Kützing) [58]. Moreover, Bulung Anggur has the closest genetic relationship distance to *Caulerpa racemosa*, at 1.200 (Table 3). Thus, it is assumed that Bulung Anggur is a species that is still understudied, given that there are few taxa in the GenBank database that have similar sequences to Bulung Anggur. 

The majority of seaweed species growing in Bali’s waters are poorly described taxonomically. These limitations can hinder various aspects, especially in cultivation and nature conservation. In this study, the application of DNA barcodes with the *tuf*A marker has successfully identified one species of *Caulerpa* seaweed in Bali. This study has been able to locate the proximity of the Bulung Boni species to other *Caulerpa* species. However, the *tuf*A marker was insufficient to distinguish Bulung Anggur accurately. Before concluding that Bulung Anggur is a novel indigenous strain, it is necessary to conduct additional research to examine its phylogeny using various markers utilized to identify Caulerpa algae, including ITS, rbcl, and rDNA [17].

## 3. Materials and Methods

### 3.1. Sample Collection

The *Caulerpa* algae, Bulung Boni and Bulung Anggur, were collected from the coastal area around Serangan Island, Bali (Figure 6). The samples were subsequently washed with clean water to remove dirt and epiphytes. Bulung Boni and Bulung Anggur were morphologically identified by referring to the morphological descriptions of several previous studies [10,24,25,59]. Fresh samples were stored in a refrigerator to be used afterward in further analysis. 

### 3.2. Maceration and GC-MS Analysis

Clean Bulung Boni and Bulung Anggur were air-dried in the shade for 2 days and then dried in the oven for 7 days at 45 °C. Dried samples were then powdered using an electric blender. The samples of each alga (15 g) were macerated in ethanol solvent (150 mL). The mixtures were kept for 96 h, then filtered and concentrated using a rotary vacuum evaporator to produce the crude extract.

The crude extract was then subjected to phytochemical analysis using GC-MS. Gas chromatography analysis was carried out on the Agilent 7890B GC (Santa Clara, CA, USA) coupled with a mass detector Agilent 5977B GC/MSD (Santa Clara, CA, USA). A measure of 1.0 µL of the extract was injected into the chromatograph at an injector temperature of 250 °C. The column (Wakosil-II5C18 4.6*200 mm, Richmond, VA, USA) oven temperature was programmed to increase from 70 °C to 290 °C at a rate of 10 °C/min. The run time for GC was 17 min. The identification of chemical compounds and the interpretation of mass spectra of GC-MS were carried out using the Wiley Spectral library.

### 3.3. DNA Extraction and PCR Amplification

Total genomic DNA of fresh Bulung Boni and Bulung Anggur samples were extracted using the Plant Genomic DNA Mini Kit (GP100, Geneaid, New Taipei City, Taiwan), following the manufacturer’s procedure. 

The *tuf*A region was amplified using published primers by Fama et al. [9], i.e., *tuf*A F 5′TGAAACAGAAMAWCGTCATTATGC-3′ and *tuf*A R 5′-CCTTCNCGAATMGCRAAWCGC-3′. PCR amplification was conducted using MyTaq HS Red Mix (Bioline, Bio-25048, London, UK). The PCR reactions were performed with the following profiles: pre-denaturation (94 °C for 2 min), 35 cycles of denaturation (98 °C for 15 s), annealing (50 °C for 30 s), extension (72 °C for 45 s), and followed by termination at 4 °C. Exactly 2 µL of PCR products were visualized using an agarose gel and selected for the sequencing process. The Big Dye Terminator Cycle Sequencing Ready Reaction Kit (Applied Biosystems, Foster City, CA, USA) was used to determine the sequence bi-directionally. 

### 3.4. Computational Analysis

Sequences obtained in this study were compared to sequences in NCBI (National Center for Biotechnology Information) by BLASTN analysis. Accessions with the highest identity percentage were selected, and multiple sequence alignment was carried out with ClustalW (MEGA 11th version, Philadelphia, PA, USA). The phylogenetic tree was constructed using the maximum likelihood method and the Tamura 3-parameter model in MEGA 11 [60]. The robustness of the tree was evaluated using bootstrapping with 1000 replicates.

## 4. Conclusions

In conclusion, the ethanolic extracts of Bulung Boni and Bulung Anggur may be a source of valuable pharmaceuticals with antioxidant, antibacterial, anticancer, anti-inflammatory, antitumor, and antiaging properties that could be manufactured as cosmetics or healthy processed foods. However, further study is necessary to elucidate other undisclosed compounds in these *Caulerpa* algae. Furthermore, this study revealed that the Balinese *Caulerpa* algae, Bulung Boni and Bulung Anggur, are genetically distinct species, with Bulung Boni being strongly connected to *Caulerpa cylindracea* and Bulung Anggur showing no close relationship to other *Caulerpa* species. Although the *tuf*A sequence was amplified effectively and generated a robust phylogenetic tree, additional research is required to compare the results obtained using different molecular markers. 

## Figures and Tables

**Figure 1 molecules-27-04879-f001:**
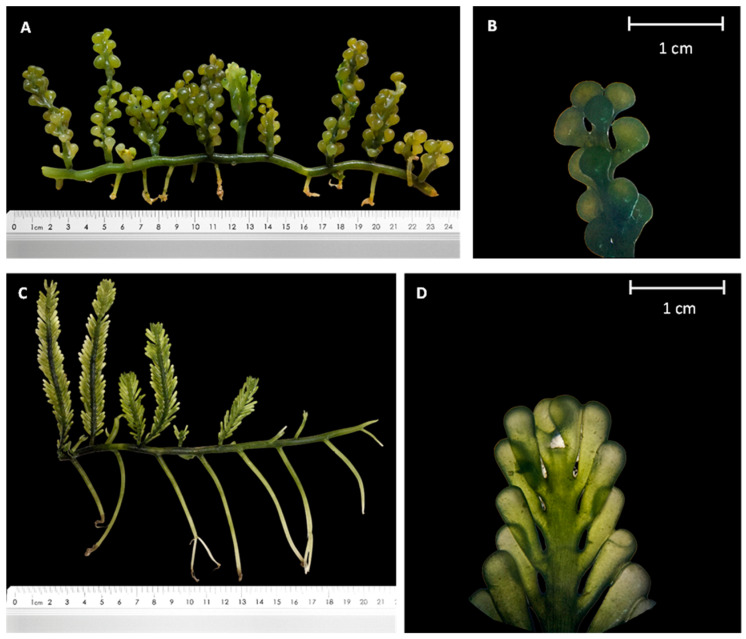
Morphological images of Bulung Anggur (**A**,**B**) and Bulung Boni (**C**,**D**).

**Figure 2 molecules-27-04879-f002:**
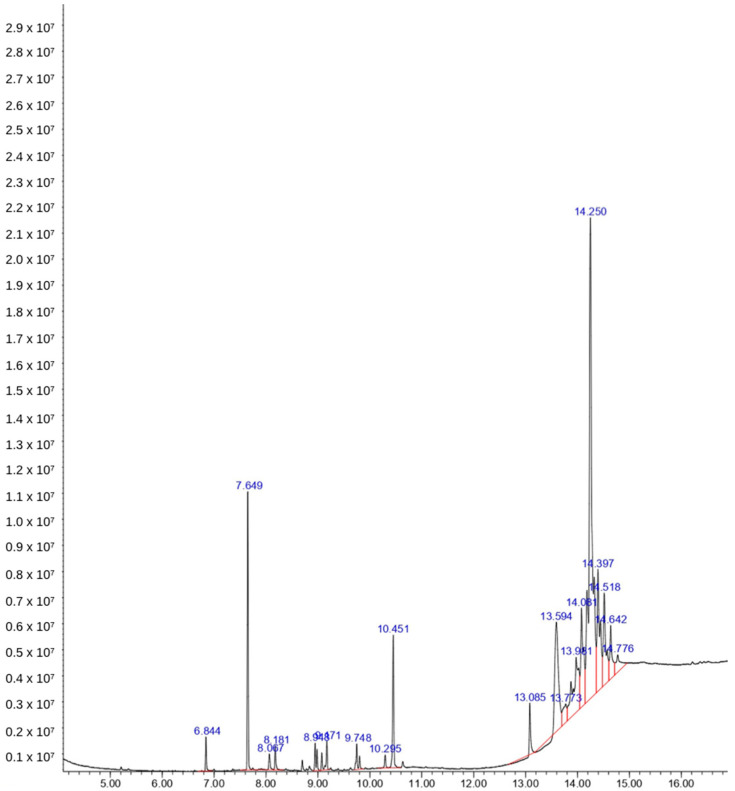
Chromatogram of Bulung Boni ethanol extract. The largest AUC showed in 14.250 of retention time, indicating that Cyclohexanamine is the dominant chemical of Bulung Boni.

**Figure 3 molecules-27-04879-f003:**
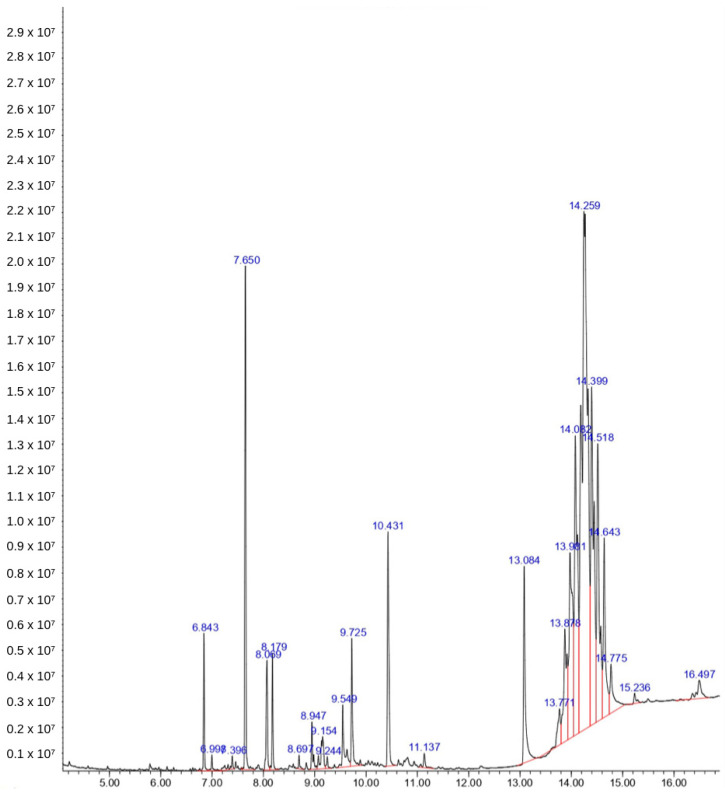
Chromatogram of Bulung Anggur ethanol extract. The largest AUC showed in 14.259 of retention time, indicating that Terephthalic acid is the dominant chemical of Bulung Boni.

**Figure 4 molecules-27-04879-f004:**
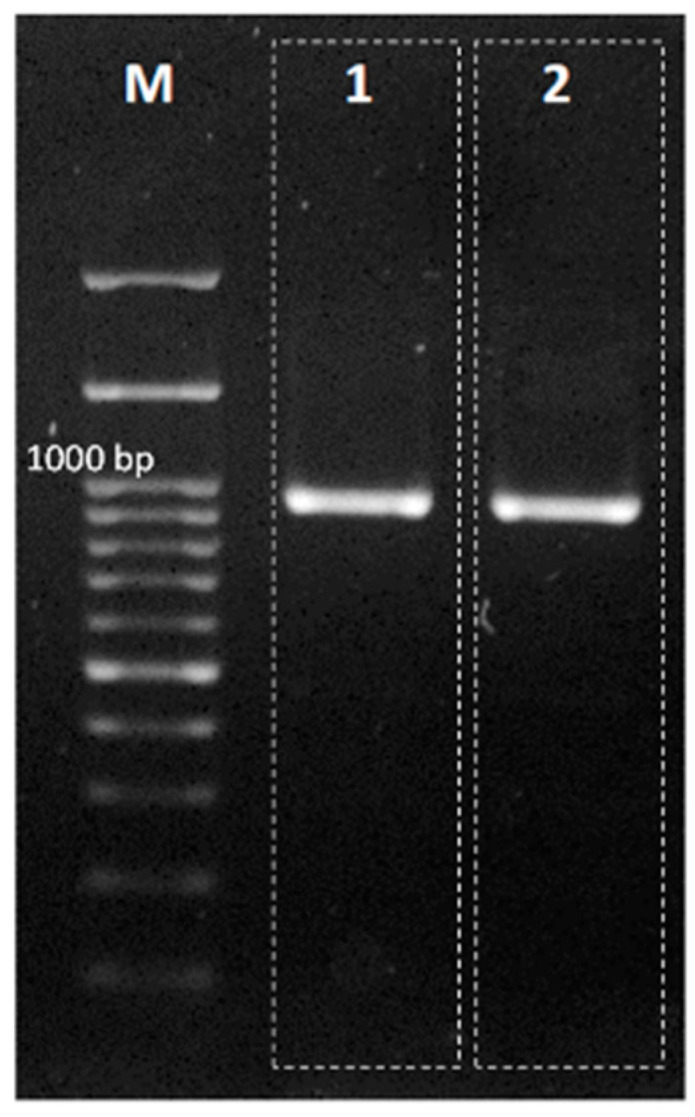
Electrophoresis results of PCR amplification of Bulung Anggur (**1**) and Bulung Boni (**2**) with *tuf*A as a primer.

**Figure 5 molecules-27-04879-f005:**
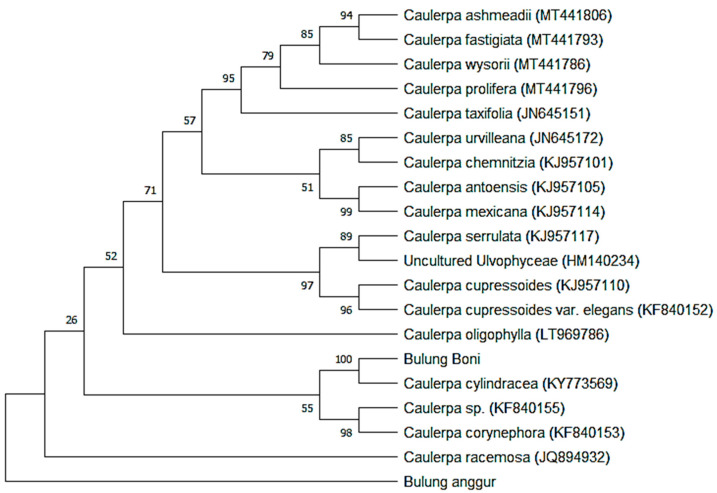
Phylogenetic tree for Bulung Boni and Bulung Anggur based on *tuf*A sequences, constructed with maximum likelihood.

**Figure 6 molecules-27-04879-f006:**
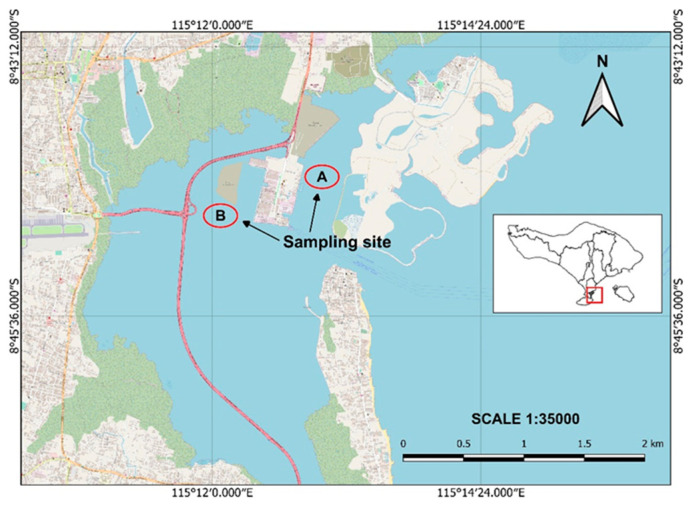
Map of sampling sites. (A: sampling site of Bulung Anggur; B: sampling site of Bulung Boni).

**Table 1 molecules-27-04879-t001:** Identified chemicals in Bulung Boni chromatogram.

Chemicals	Rt	AUC	Molecular Weight	Formula
Cyclododecane	6.844	0.78	168.32	C_12_H_24_
Pentanoic acid	7.649	6.48	102.13	C_5_H_10_O_2_
3-Heptadecene	8.067	0.69	238.5	C_17_H_34_
Eicosane	8.181	0.65	282.5	C_20_H_42_
Neophytadiene	8.948	1.08	278.5	C_20_H_38_
11,13-Dimethyl-12-tetradecen-1-ol acetate	9.748	1.09	282.5	C_18_H_34_O_2_
1-Dodecanol	10.451	3.92	186.33	C_12_H_26_O
Androst-5,15-dien-3ol acetate	13.594	9.03	314.5	C_21_H_30_O_2_
2-Naphthalene-sulfonic acid	13.981	7.89	924.9	C_39_H_31_N_7_NaO_13_S_3_^+^
Cyclohexanamine	14.250	36.91	99.17	C_7_H_15_N
2-p-Nitrophenyl-1,3,4-Oxadiazol-5-one	14.776	2.04	207.14	C_8_H_5_N_3_O_4_

AUC, Area Under Curve (%); Rt, retention time. Molecular weights are expressed in g/mol. All chemicals are listed from shortest to longest Rt.

**Table 2 molecules-27-04879-t002:** Identified chemicals in Bulung Anggur chromatogram.

Chemicals	Rt	AUC	Molecular Weight	Formula
1-Decene	6.843	1.24	140.27	C_10_H_20_
Pentadecene	6.998	0.23	210.4	C_15_H_30_
Dodecane	7.396	0.35	170.33	C_12_H_26_
8-Heptadecene	8.069	1.58	238.5	C_17_H_34_
Heptadecane	8.179	1.10	240.5	C_17_H_36_
Loliolide	8.697	0.16	196.24	C_11_H_16_O_3_
Neophytadiene	8.947	0.65	278.5	C_20_H_38_
1-Nonadecene	9.154	0.93	266.5	C_19_H_38_
Hexadecanoic acid	9.549	1.29	256.42	C_16_H_32_O
Phytol	10.431	3.53	296.5	C_20_H_40_O
1-Hydroxy-1-o-fluorophenyl-4-nitroimidazole-3-oxide	13.771	0.90	100.08	C_4_H_5_N_3_O_2_
Terephthalic acid	14.259	31.99	166.13	C_6_H_4_(CO_2_H)_2_
1,2-Cyclohexanedicarboxylic acid	14.399	10.69	172.18	C_8_H_12_O_4_

AUC, Area Under Curve (%); Rt, retention time. Molecular weights are expressed in g/mol All chemicals are listed from shortest to longest Rt.

**Table 3 molecules-27-04879-t003:** Barcode alignment for Bulung Boni and Bulung Anggur collected from Bali. The best matched identify with NCBI-BLAST are noted with their GenBank accession number, pairwise distance, and similarity percentage (ID).

Species	Accession No	Bulung Boni		Bulung Anggur	
		Pairwise	ID	Pairwise	ID
*Caulerpa racemose*	JQ894932	0.013	94.10%	1.200	39.60%
*Caulerpa cylindracea*	KY773569	0.000	94.30%	1.250	40.80%
*Caulerpa sp.*	KF840155	0.010	97.70%	1.250	41.00%
*Caulerpa corynephora*	KF840153	0.009	96.40%	1.226	40.20%
*Caulerpa serrulate*	KJ957117	0.026	95.60%	1.259	41.10%
*Caulerpa cupressoides*	KJ957110	0.027	95.40%	1.262	41.00%
*Caulerpa cupressoides var. elegans*	KF840152	0.028	95.20%	1.240	40.20%
*Caulerpa urvilleana*	JN645172	0.026	95.80%	1.246	41.20%
Ulcultured *Unvophyceae*	HM140234	0.027	91.80%	1.269	40.40%
*Caulerpa antoensis*	KJ957105	0.035	94.70%	1.275	41.00%
*Caulerpa mexicana*	KJ957114	0.034	94.90%	1.257	41.20%
*Caulerpa chemnitzia*	KJ957101	0.035	94.70%	1.250	41.40%
*Caulerpa wysorii*	MT441786	0.037	88.20%	1.289	40.30%
*Caulerpa prolifera*	MT441796	0.038	95.00%	1.301	40.30%
*Caulerpa ashmeadii*	MT441806	0.039	94.70%	1.241	40.60%
*Caulerpa oligophylla*	LT969786	0.021	87.90%	1.226	37.00%
*Caulerpa fastigiate*	MT441793	0.041	92.30%	1.234	40.30%
*Caulerpa taxifolia*	JN645151	0.044	94.20%	1.278	40.60%
Bulung boni	-	-		1.229	41.10%
Bulung anggur	-	1.229	41.10%	-	-

Pairwise, pairwise distance; ID, percentage of identity.

## Data Availability

This study uses data published in GenBank, which can be accessed on https://www.ncbi.nlm.nih.gov/ (accessed on 9 December 2021). These data are used in the BLAST application to compare the sequences in the GenBank database with the sequences obtained in this study.

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
