# Peer review of "Phytochemical Analysis and Molecular Identification of Green Macroalgae Caulerpa spp. from Bali, Indonesia"

_molecules, 2022, doi:10.3390/molecules27154879_

Round 1

Reviewer 1 Report

The work under the title "Phytochemical Analysis and Molecular Identification of Green Macroalge Caulerpa spp. From Bali, Indonesi"a I found an interesting. In the introduction, it was pointed out that the chemical composition of Caulerpa algae, particularly the Bulung Boni and Bulung Anggur species, has not yet been studied. In this context, the authors identified the bioactive compounds in these species and proposed their further use based on their composition. Thus, the main objective of this research is obvious and well elaborated.

The results in tables and figures are well presented, so I have no objection to their presentation. The results are adequately discussed and compared with other works. The conclusion confirms the obtained results and the references inserted in the main document show a good connection between this investigation and other works.

The material and methods, as well as results and discussion part are understandable. The article is good from grammatical and structural point of view, and for my perspective is acceptable for publication in this journal.

Minor errors:

Introduction

Line 47: Please insert the reference number after Julyasih et al. See also line 59. Line 109: Please explain how the BOD/Biological Oxygen Demand) is measured. Concentration (2,8-5,4 mg/L)?

Reviewer 2 Report

Hereafter are reported the comments arising from my revision of the manuscript entitled "Phytochemical Analysis and Molecular Identification of Green Macroalgae Caulerpa spp. from Bali, Indonesia".  

I have read at great length the work. I found this article informative in regards to background information.

The manuscript appears to have sufficient scientific quality and may be of interest to the readers of the Journal. 

The scientific value of the manuscript is good (although not outstanding)

The manuscript brings some new information. 

Abstract: first two sentences need to rewrite

Introduction; good

Materials and methods: no statistical analysis but in general good.

Results : good

Discussion: good
